# Raman and Photoluminescence Studies of Quasiparticles in van der Waals Materials

**DOI:** 10.3390/nano15020101

**Published:** 2025-01-10

**Authors:** Mansour M. AL-Makeen, Mario H. Biack, Xiao Guo, Haipeng Xie, Han Huang

**Affiliations:** 1Hunan Key Laboratory of Super-Microstructure and Ultrafast Process, School of Physics, Central South University, Changsha 410083, China; almakeen.mansour@csu.edu.cn (M.M.A.-M.); 222218017@csu.edu.cn (M.H.B.); gzguoxiao163@163.com (X.G.); 2Physics Department, Almahweet University, Almahweet 36080, Yemen; 3School of Physical Science and Technology, Xinjiang University, Urumqi 830046, China

**Keywords:** many-body effect, phonon anharmonicity, electron–phonon coupling, exciton, anisotropy

## Abstract

Two-dimensional (2D) layered materials have received much attention due to the unique properties stemming from their van der Waals (vdW) interactions, quantum confinement, and many-body interactions of quasi-particles, which drive their exotic optical and electronic properties, making them critical in many applications. Here, we review our past years’ findings, focusing on many-body interactions in 2D layered materials, including phonon anharmonicity, electron–phonon coupling (*e-ph*), exciton dynamics, and phonon anisotropy based on temperature (polarization)-dependent Raman spectroscopy and Photoluminescence (PL). Our review sheds light on the role of quasi-particles in tuning the material properties, which could help optimize 2D materials for future applications in electronic and optoelectronic devices.

## 1. Introduction

Atomically thin 2D layered materials are typically composed of a single layer or just a few layers of atoms that display distinctive electrical, optical, and mechanical characteristics, offering significant potential in technology applications [1]. For example, graphene is the archetypal 2D material renowned for its high charge carrier mobility, chemical stability, and outstanding mechanical strength [2]. The unique characteristics of 2D materials stem from their layered structure held together by vdW interactions, enabling innovative applications in electronics, photonics, and sensing technologies, for instance, transition metal dichalcogenides (TMDs), black phosphorus, and several other layered compounds that have been investigated for use in flexible electronics, high-performance transistors, and smart sensors [3,4,5]. Two-dimensional materials exhibit a diverse array of electronic states that significantly diverge from those of their bulk equivalents, attributed to quantum confinement and intensified many-body interactions, leading to emergent phenomena involving quasi-particles like *e-ph* and phonon–phonon (*ph-ph*) interactions as well as excitons, etc., that dominate their optical and electronic properties [6,7].

Various methods are employed to investigate quasi-particles in 2D vdW materials, providing insights into their electronic and optical properties. For instance, Scanning tunneling spectroscopy (STS) offers a distinct approach for investigating the interaction between individual electrons (holes) and lattice phonons [8]. Angle-resolved photoemission spectroscopy (ARPS) provides an effective method to describe the many-body interactions between quasi-particles at the Fermi level [9,10]. Similarly, transport measurements (e.g., thermoelectric transport, steady-state thermal techniques, etc.) are used to investigate quasi-particle dynamics, such as carrier mobility, transport coefficients, and scattering mechanisms in vdW materials [11,12].

One of the quick and nondestructive characterization and analysis techniques with exceptional spatial and spectral resolution, Raman spectroscopy stands out and is relevant at both laboratory and mass-production levels [13,14,15]. The common characteristics shared by the Raman peaks that reflect the lattice vibrations (phonons) in 2D materials are their line shape, peak position, full width at half maximum (FWHM), and intensity (I), which provide valuable insight into the 2D materials’ physical and chemical properties, including quantum interference, electronic states, phonon frequency, many-body interaction effects of quasi-particles such as temperature dependence for *e-ph* coupling, phonon anharmonicity, thermal expansion, etc. [13,16,17,18]. In addition, polarized Raman spectroscopy can measure the Raman scattering intensity depending on the polarization direction of both the incident and scattered light, providing insight into the phonon symmetry, which is sensitive to the crystal structure, strain, and electronic properties for understanding the crystallographic orientation of 2D materials [19,20]. Photoluminescence (PL) is another important tool that serves as an essential technique for investigating 2D materials, offering additional insights alongside Raman spectroscopy [21]. It is sensitive to altering the 2D materials’ electronic band structure, reflecting the variations caused by layer number, strain, and defects, which are crucial for comprehending many-body effects, such as exciton dynamics and recombination processes [22,23,24]. PL measurements can detect the anisotropy in the optical properties, which is associated with the fundamental crystal structure and the electronic interactions present within the material; such properties are crucial in spintronics and optoelectronics applications, where the exploitation of the directional properties plays a significant role [21,25,26].

Quasi-particles have a key role in how materials behave at the microscopic level, which affects their macroscopic characteristics and technological uses. Their importance in simplifying complicated many-body interactions in materials enables the modeling and comprehension of macroscopic features such as conductivity, thermal behavior, and optical responses. [27]. The *e-ph* coupling plays a crucial role in the formation of Cooper pairs within superconductors, significantly influencing their transition temperatures [28]. It can affect the materials’ thermal and electrical conductivity, and thus their transport properties [29]. The anharmonic phonon interactions result in temperature-dependent alterations in the lattice vibrations, phase transitions, and the stability of materials, considerably influencing the thermal expansion and heat capacity [30,31,32]. Therefore, understanding their decay processes is important in practical applications. Excitons have distinctive characteristics in 2D materials owing to robust electron–hole (*e-h*) correlations [33]. Comprehending these excitonic states is essential for the advancement of optoelectronic applications [34].

In this review, we summarize our past years’ works, highlighting the many-body interaction effects based Raman spectroscopy and PL techniques. We specifically focus on the phonon anharmonicity in layered vdW materials and their related *e-ph* coupling based on temperature-dependent Raman spectroscopy. The exciton dynamics based on PL are discussed. Furthermore, we discuss the phonon anisotropy in vdW materials and heterostructures and their dependence on excitation wavelengths according to polarized Raman scattering.

## 2. Many-Body Interaction Effects Based on Raman Spectroscopy and PL

Raman and PL spectroscopy are effective techniques for examining many-body interactions of quasi-particles in vdW materials, including phonon anharmonicity, *e-ph* coupling, excitons, and phonon anisotropy, etc. Before delving through the details, we have summarized the Raman bands of the materials addressed in this review in Table 1 to provide readers with a clear overview.

### 2.1. Phonon Anharmonicity

Phonon anharmonicity is essential for comprehending lattice dynamics, especially in investigating materials’ thermal characteristics and their responses to temperature fluctuations. Indeed, harmonic or quasi-harmonic approximations are often used to study the vibrational characteristics of nano-structures at a temperature close to 0 K while ignoring the quantum properties of the nuclei of ions. In this scenario, the harmonic phonons are treated as non-interacting and have an unlimited lifetime and then infinite thermal conductivity [35]. This approximation could not effectively capture the physical properties of the lattice (dynamically unstable harmonic levels) at finite temperatures. The quasi-harmonic approximation is an extension of the harmonic approximation used in solid-state physics to investigate lattice vibrations (phonons). Unlike simple harmonic approximation, in which atoms oscillate around their equilibrium positions with no interaction changes, quasi-harmonic approximation takes into account the volume dependence of phonon frequencies, allowing for the study of thermal expansion and temperature-dependent material properties, which harmonic approximation fails to describe accurately. In realistic materials, the potential energy of atomic bonds is not exclusively harmonic, resulting in departure from ideal behavior. These aberrations, named phonon anharmonicity, are crucial for elucidating thermal expansion, *ph-ph* scattering, and the subsequent decay of phonons [36]. Phonon anharmonicity significantly influences a material’s thermal conductivity and stability. Raman spectroscopy can be used to reveal vibrational modes (phonons) in materials. In this context, the anharmonicity is demonstrated by temperature-dependent shifts in phonon frequencies and broadening in linewidths. The temperature dependence elucidates phonon decay mechanisms and lifetimes, both affected by anharmonic interactions. The fundamental process of phonon anharmonicity encompasses higher-order *ph-ph* interactions, wherein phonons can interact with and decay into other phonons. Higher-order terms generally describe this interaction in crystal potential energy, particularly the third and fourth orders. The three-phonon (*3-ph*) decay process includes the decay of one phonon into two phonons or the merging of two phonons into one phonon at a finite time. The four-phonon (*4-ph*) process includes one phonon decaying into three phonons, or vice versa, or two phonons decaying into two new phonons (two new states) [37,38].

According to the principles of classical mechanics, the oscillation frequency for generic nonlinear systems is primarily dependent on the oscillation amplitude [39]. Hence, as the temperature increases, average oscillation amplitudes expand, leading to the anharmonicity that ultimately results in temperature-dependent phonon spectra. A similar effect could be explained in quantum mechanics by considering terms of phonon self-energy caused by *ph-ph* interactions [40]. Phonon self-energy is described as Δ(ω)+iΓ(ω), in which Δ(ω) is the real term related to the frequency shift of phonons that occurs as a result of scattering by other phonons and to the temperature dependency of the phonon frequency. The Γ(ω) represents the imaginary part that expresses the phonon decay probability (damping process) and refers to the phonon lifetime inverse (1/τ). The phonon self-energy (real part) includes *3-* and *4-ph* scattering processes to determine frequency shift ($∆\omega_{anhr}$). However, a quasi-harmonic (thermal expansion) ($∆\omega_{V}$) contribution should be included because of the temperature dependence of volume V(T). On the other hand, the frequency shift is determined by three contributions: the *3-ph*, *4-ph*, and quasi-harmonic effects. Therefore, the frequency shift of the phonon mode as a function of temperature can be expressed according to the Balkanski model by the following equation [41];(1)$$∆\omega =\omega \left(T\right)-\omega_{0}=∆\omega_{V}+∆\omega_{anhr}$$
where $\omega_{0}$ is the phonon mode frequency at 0 K. The contribution of $∆\omega_{V}$ can be determined using the Grüneisen model as follows [42];(2)$$∆\omega_{V}=\omega_{0}\left\{exp\left(-n\gamma \int_{0}^{T}\alpha \left(T\right)dT\right)-1\right\}$$
where *n* refers to the degeneration of the Raman active modes, α(T) is the thermal expansion coefficient, and γ represents the Grüneisen parameter.

The contribution from *3-* and *4-ph* decay processes can by determined using the equation;(3)$$∆\omega_{anhr}=A\left(1+\frac{2}{e^{x}-1}\right)+B\left(1+\frac{3}{e^{y}-1}+\frac{3}{{{(e}^{y}-1)}^{2}}\right)$$
where *A* and *B* represent the constants fitting of *3-* and *4-ph* decay processes, respectively. $x=\hbar \omega_{0}/2k_{B}T$, $y=\hbar \omega_{0}/3k_{B}T$, in which ℏ and $k_{B\;}$ are the reduced Planck and Boltzmann constants, respectively.

The linewidth can be determined according to the following equation, including *3-* and *4-ph* contributions [18];(4)$$\Gamma \left(T\right)=\Gamma_{0}+\Gamma_{anhr}(T)$$
where $\Gamma_{0}$ denotes the temperature-independent constant resulting from the limited resolution of the spectrometer or imperfections present in real crystallographic samples. $\Gamma_{anhr}(T)$ represents the anharmonic linewidth contribution (*3-* and *4-ph*) and can be expressed as;(5)$$\Gamma_{anhr}(T)=C\left(1+\frac{2}{e^{x}-1}\right)+D\left(1+\frac{3}{e^{y}-1}+\frac{3}{{{(e}^{y}-1)}^{2}}\right)$$

In which *C*, and *D* represent the anharmonic linewidth constants.

Raman spectroscopy is useful for investigating thermal phonon behavior in low-dimensional materials, such as 7-AGNRs, graphene, etc. Figure 1a,b represent STM images of long-range-ordered armchair chains of DBBA molecules self-assembled at room temperature (RT) on Au (111) substrate (for synthesizing 7-AGNRs) and a high-quality 7-AGNR sample, respectively [43]. The typical RT Raman spectrum of 7-AGNR is shown in Figure 1c. Graphene is represented by dominated G and 2D vibrational modes, while 7-AGNRs have more Raman active modes arising from spatial confinement effects; more details have been previously reported [44,45]. The strong peak represents the G vibration mode centered at 1595 cm^−1^, which arises from the in-plane C-C stretching. The peak at 1350 cm^−1^ is ascribed to the 2D mode of defective graphene; however, in this spectrum, the vibration modes at around 1220 cm^−1^ (CH_1_ mode), 1255 cm^−1^ (CH_2_ mode), and 1340 cm^−1^ (D mode) arise not from defects but from the disruption of the periodicity of an ideal honeycomb lattice due to the edges being hydrogen passivated, leading to C-H bending [46]. 7-AGNR exhibits unique vibrations at low frequencies, potentially reflecting the inherent features, e.g., the peak center at 396 cm^−1^ is ascribed to a radial breathing-like mode (RBLM), which arises from the relative motions between two different components of the 7-AGNRs. The mode at 953 cm^−1^ is attributed to the third overtone mode of the RBLM (RBLM3), exhibiting properties analogous to those of the RBLM. All these modes show high symmetric lineshapes, and can be fitting by Lorentzian function.

Figure 1d represents the Raman shift of the G mode of 7-AGNRs as a function of temperature in the range from 80 to 520 K, exhibiting a linear redshift with increased temperature (green triangles), and similarly for the D mode of 7-AGNRs, represented by red squares. The blue dots data represent the frequency shift of the G mode of single-layer graphene (SLG) grown by CVD on Cu foil for comparison, showing linear redshift with increased temperature from 80 to 440 K (see Figure 1d), and then turn to blueshift due to the thermal expansion coefficient mismatch resulted slip [47]. Similar trends are observed for C-H modes [43]. All linear Raman shift of these modes can be fitted by Grüneisen expression (as represented by solid lines in Figure 1d) [48];(6)$$\omega =\omega_{0}+\chi T$$
where *χ* is the first-order temperature coefficient. Therefore, it is calculated to be $\chi_{G\;mode}$ (7-AGNRs) $=-2.64\times 10^{-2}$ cm^−1^ K^−1^, $\chi_{D\;mode}$ (7-AGNRs) $=-3.07\times 10^{-2}\;$cm^−1^ K^−1^, and $\chi_{G\;mode}$ (SLG/Cu) $=-5.62\times 10^{-2}$ cm^−1^ K^−1^. The $\chi_{G\;mode}$ of 7-AGNRs is larger than the $\chi_{G\;mode}$ of SLG, and even larger than those of multi- and single-walled carbon nanotubes (CNTs) ($-2.8\times 10^{-2}$, $-3.2\times 10^{-2}$ cm^−1^ K^−1^) [49,50], indicating that 7-AGNRs have much better thermal stability compared with CNTs; this may be ascribed to the structural configurations of CNTs, which elevate stress and diminish heat stability. The extracted $\omega_{0}$ values of the G modes of 7-AGNRs and SLG are 1612.0 and 1602.2 cm^−1^, respectively, where the larger value for 7-AGNRs compared with SLG suggests that the 7-AGNRs experience less strain. Given that 7-AGNRs are large molecules in comparison with SLG, the interfacial strain may be alleviated through molecular slipping on the surface. Thus, the 7-AGNRs–substrate interaction effect on the temperature dependence can be disregarded.

Figure 1e demonstrates the temperature dependence of the linewidth of the G mode for 7-AGNRs, showing nonlinear broadening with increased temperature, and can be fitting well using the quadratic polynomial function as follows;(7)$$\Gamma =\Gamma_{0}+\left(\frac{\partial \Gamma }{\partial T}\right)\Delta T+\left(\frac{\partial^{2}\Gamma }{{\partial T}^{2}}\right){∆T}^{2}$$

In which $\frac{\partial \Gamma }{\partial T}$ and $\frac{\partial^{2}\Gamma }{{\partial T}^{2}}$ represent the first and quadratic temperature coefficient of linewidth for the G mode of 7-AGNRs.

The temperature dependence of linewidth comes from the competition between *ph-ph* interaction and *e-ph* coupling. This can be manifested by the behavior of the linewidth curve before turning at 180 K (*e-ph* contribution) and over 180 K (*ph-ph* interaction dominated) in Figure 1e.

Different from the G mode, the frequency shift of the RBLM in Figure 1f shows nonlinear redshift dependence with increased temperature. Indeed, the nonlinear frequency shift at different temperatures reflected three contributions, $∆\omega_{anhr}$, $∆\omega_{V}$, and the thermal expansion coefficient mismatch of the 7-AGNRs–substrate interaction. The latter contribution can be ignored because of weak interfacial interaction, as discussed above. In addition, the contribution of $∆\omega_{V}$ could be ignored, similar to previous reports of the G mode in multilayer (ML) G and MLGNRs that suggest it plays a minor role [51,52]. Therefore, the frequency shift of the RBLM can be only described by the multi-phonon decay processes, including *3-* and *4-ph,* using the equation (Equation (3)). The best fitting in Figure 1f shows that the *3-ph* contribution gives rise to blueshift far from experimental data. In contrast, *4-ph* gives nonlinear redshift behavior to the RBLM with increased temperature (predominant). On the other hand, the *4-ph* decay process drives the frequency shift of the RBLM nonlinearly redshift with increased temperature [43]. Figure 1g represents the linewidth of the RBLM as a function of temperature, showing nonlinear broadening. It is known that in a defect-free sample the linewidth behavior arises from phonon decay to lower energy by *ph-ph* interactions resulting from anharmonicity in the lattice vibrations or producing (*e-h*) pairs due to *e-ph* coupling. The contribution of the latter can be disregarded here because it is valid for zero bandgap systems, such as graphene and graphite [18], while 7-AGNRs have a bandgap of ~2.3 eV. Thus, the linewidth of the RBLM can only be described by the *ph-ph* decay process presented in Equations (4) and (5), revealing best fitting, dominated by the *4-ph* decay process, which gives nonlinear broadening for linewidth with increased temperature.

It is important to mention that the phonon thermal behavior dependent on Raman spectroscopy investigation can give us deep insight into the material–substrate interaction, even with increased layers stacking, which is important in optical and electronic applications [53,54,55]. In this context, Figure 1h represents the Raman shift frequency of the G mode of SL (black squares), bi-layer (BL) (red dots) and tri-layer (TL) (blue triangles) graphene on SiO_2_/Si substrate taken from OM inserted in panel (h) at different temperatures [56]. The extracted $\chi_{G\;mode}$ (SLG/SiO_2_/Si) $is-3.43\times 10^{-2}$ cm^−1^ K^−1^, smaller than that for SLG/Cu, due to the effect of different substrates [57]. $\chi_{G\;mode}$ (BLG) $=-2.67\times 10^{-2}$ cm^−1^ K^−1^, and $\chi_{G\;mode}$ (TLG) $=-2.21\times 10^{-2}$ cm^−1^ K^−1^, respectively. Actually, the total first temperature coefficient (χ) comes from different contributions, such as the *ph-ph* interaction, thermal expansion ($\chi_{V}$), and substrate effect ($\chi_{sub}$). The anharmonic *ph-ph* interaction is independent of the layers stacking in the first temperature coefficient, and it is disregarded in this context because the synthesized graphene is A–B stacked graphene. Thus, the phonon dispersion and the *ph-ph* channels in the different layers are similar, excluding the ZA channel, which decouples with the decay channel in an anharmonic *ph-ph* interaction [58]. Moreover, the contribution of thermal expansion is also ignored here according to the discussion above [51,52]. Therefore, the total first temperature coefficient (χ) is only driven by the substrate effect ($\chi_{sub}$), as explained by Equation (5) in the reference [56], displaying a linear decreasing trend with an increase in the number of layers, which is ascribed to the diminished substrate influence on phonon behavior.

**Figure 1 nanomaterials-15-00101-f001:**
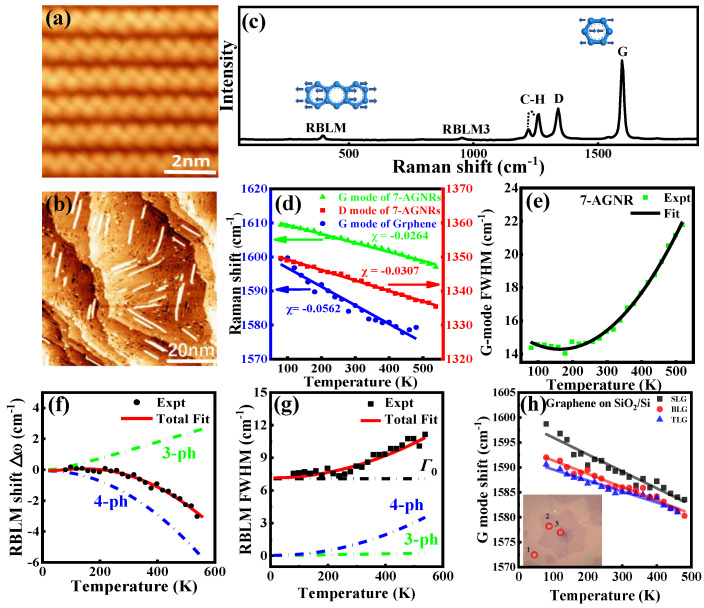
Typical STM images of long-order self-assembled DBBA molecular chains (**a**) and optimized 7-AGNR sample (**b**). (**c**) Raman spectrum at RT of 7-AGNR. Insert the vibration modes displacement. (**d**) Temperature dependence of position shift for G and D phonon modes of 7-AGNR, compared with G phonon mode for grown graphene/Cu using CVD (highlighted by blue color), fitted by Equation (6). (**e**) Linewidth as a function of temperature for 7-AGNR G phonon mode, fitted using Equation (7). (**f**,**g**) represent 7-AGNR RBLM phonon position shift and linewidth evolution with temperature, fitted by Equations (3)–(5), respectively. Reproduced with permission from [43], copyright 2022, Elsevier. (**h**) Temperature dependence of position shift of G mode for SL, BL, and TL graphene/SiO_2_/Si, respectively. Reproduced with permission from [56], copyright 2022, Springer Nature.

Figure 2a demonstrates the AFM image of high-quality α-MoO_3_ flakes grown by physical vapor deposition (PVD), showing a layered structure with a step height of 1.4 nm [59]. Inset shows the OM of the centimeter-scale α-MoO_3_ flakes. The typical SEM image in Figure 2b represents the sample of α-MoO_3_ flakes with a smooth surface and long straight edges. Figure 2c displays the typical RT Raman spectra of the powder and grown crystal of α-MoO_3_, showing three phonon modes at high-frequency band centered around 994.5, 818.9, and 666.4 cm^−1^, assigned to Mo–O stretching vibration phonon modes (SVPMs) I, II, and III, respectively. The other vibrational modes, located below 500 cm^−1^, belong to the bending vibrations and translations of rigid MoO_4_. Figure 2d represents the color mapping of temperature dependence of SVPM I, showing nonlinear behavior with increased temperature. The corresponding frequency shift of SVPM I as a function of temperature in Figure 2e demonstrates anomaly nonlinear shifting with temperature, blueshifted up to ~340 K, then redshifted with temperature. The best fitting according to Equations (1)–(3) reveals that the frequency shift of SVPM I mainly results from the competition between the thermal expansion (redshifted) and *4-ph* decay process (blueshifted), represented separately by blue and black dashed lines, respectively; while the *3-ph* decay process causes a minor frequency shift (green dashed line). The substrate effect contribution can be considered for monolayer or a few layers of materials [60], while it is disregarded here for α-MoO_3_ flakes with a thickness of about 120 nm. The linewidth of SVPM I nonlinearly broadens with increased temperature, as shown in Figure 2f, described well according to the *3-* and *4-ph* decay processes and $\Gamma_{0}$ contribution by Equations (4) and (5). Quantitatively, the linewidth arises from the competition between the *3-* and *4-ph* decay processes at low temperatures, and then it is dominated by the *4-ph* decay process with increased temperature. Figure 2g shows the color mapping evolution of SVPM II with clear broadening. The abnormal nonlinear frequency shift of SVPM II in Figure 2h is described similarly to SVPM I, dominated by the competition between thermal expansion and *4-ph* contributions. The *3-ph* decay process causes a slightly linear broadening on the linewidth of SVPM II with increased temperature, while the *4-ph* decay process dominates the entire *ph-ph* scattering mechanism, as shown in Figure 2i. In contrast to SVPMs, the low-frequency phonon modes show different frequency shifts with temperature. For example, the phonon mode at 156 cm^−1^ displays linear redshift and nonlinear linewidth broadening with increased temperature, and both are dominated by the competition between the *3-* and *4-ph* decay processes; more details in the reference [59].

The typical AFM image in Figure 3a represents the layered exfoliated b-As nanoflake of 64 nm thickness. Figure 3b shows the corresponding polarized Raman spectra (at RT) at different incident polarization angles of 0°, 45°, and 90°, respectively, revealing three Raman active modes, $A_{g}^{1}$ (221 cm^−1^), *B*_2g_ (227 cm^−1^), and $A_{g}^{2}$ (255 cm^−1^). *B*_2g_ shows strong intensity at θ = 45° and weak intensity at θ = 0° and 90°, while $A_{g}^{1}$ and $A_{g}^{2}$ display strong intensity at θ = 0° [61]. The temperature dependence of the $A_{g}^{1}$ and $A_{g}^{2}$ modes is performed at the temperature range 80–300 K in the armchair (AC) direction, where the *B*_2g_ mode is hard to observe. Both frequency shifts of $A_{g}^{1}$ and $A_{g}^{2}$ modes in Figure 3c,d show nonlinear redshifts, dominated mainly by the *3-ph* decay process at low temperatures and the competition between the *3-* and *4-ph* decay processes at high temperatures (as represented by the red and blue dashed lines), according to the best fitting by Equations (1) and (3), considering only the contribution from the *ph-ph* decay processes. The substrate effect and thermal expansion coefficient are ignored due to the flake thickness of 64 nm, as well as the fact that the thermal expansion has a minor effect on the exfoliated b-As nanoflake similar to BP [62]. The linewidths broadenings of both $A_{g}^{1}$ and $A_{g}^{2}$ modes in Figure 3e,f exhibit nonlinear behaviors with increased temperature. The fitting curves (using Equations (4) and (5)) indicate that the *3-ph* decay process predominates linearly up to 175 K; afterward, the competition between the *3-ph* and *4-ph* decay processes becomes significant, resulting in nonlinearity broadening behavior, as represented by the red (*3-ph*) and blue (*4-ph*) dashed lines.

### 2.2. Electron–Phonon Coupling

The *e-ph* coupling is one of the basic interactions among quasi-particles in solids, playing a significant function in several physical phenomena [63,64]. Especially in metals, low-energy electronic excitations undergo significant modifications due to their interaction with lattice vibrations (phonons), which affect transport and thermodynamic characteristics. It is an essential mechanism underlying conventional superconductivity, as explained by the Bardeen–Cooper–Schrieffer (BCS) theory [65]. The *e-ph* coupling promotes superconductivity and charge density wave (CDW) phenomena [28,66,67]. In the context of Raman spectroscopy, the *e-ph* coupling can be manifest in the phonon linewidth changes with temperature [68,69].

The *e-ph* coupling contribution on the linewidth broadening, $\Gamma^{e-ph}(T)$, can be described according to the Fermi golden rule [70], in which the electronic state $|\left.ki\right\rangle$ with vector k is excited by a phonon with wave vector q to the state $|\left.\left(k+q\right)j\right\rangle$ during the process of creating an *e-h*. Therefore $\Gamma^{e-ph}(T)$ can be written as;(8)$$\Gamma_{q}^{e-ph}(T)=\frac{4\pi }{N_{k}}\sum_{k,i,j}{\left|g_{\left(k+q\right)j,\;ki}\right|}^{2}\left[f_{ki}\left(T\right)-f_{\left(k+q\right)j}(T)\right]\times \delta \left[\epsilon_{ki}-\epsilon_{\left(k+q\right)j}+\hbar \omega_{q}\right]$$
where $g_{\left(k+q\right)j,ki}$ refers to the *e-ph* matrix elements, and $\omega_{q}$ is the phonon frequency. The number of k vectors is denoted by $N_{k}$, and the $f_{ki}\left(T\right)$ indicates the Fermi–Dirac occupation at a given temperature, *T*, for the electron state with energy $\epsilon_{ki}$. δ represents the Dirac delta distribution. This model simplifies to Equation (9) below according to references [18,71] by considering the energy conservation condition $\epsilon_{ki}+\hbar \omega_{q}=\epsilon_{\left(k+q\right)j}$, which determines the electron states that contribute to the sum in Equation (8). In addition, the ki state is occupied and the $\left(k+q\right)j$ is empty, and thereby the term $\left[f_{ki}\left(T\right)-f_{\left(k+q\right)j}(T)\right]\neq 0.$ Therefore, the electrons close to the Fermi level contribute to the *e-ph* coupling [18];(9)$$\Gamma^{e-ph}\left(T\right)=\Gamma^{e-ph}\left(0\right)\left(\frac{1}{e^{-x}+1}-\frac{1}{e^{x}+1}\right)$$
where ${\Gamma_{i}}^{e-ph}\left(0\right)$ indicates the linewidth of the vibration mode at low temperature (0 K).

It’s known that the layered PdTe_2_ is recognized for its inherent topological conventional superconductivity attributed to enhanced *e-ph* coupling [66,72]. Liu et al. [73] reported a study into the temperature dependence of the ultrafast carrier and phonon dynamics in PdTe_2_, revealing two different carrier relaxation processes (fast *τ*_*f*_, and slow *τ*_*s*_) with varying scales of time, which originated from the *e-ph* thermalization (*τ*_*f*_, at sub-picosecond time scale) and phonon-assisted *e-h* recombination (*τ*_*s*_, at ~7–9.5 ps), respectively. Recently we successfully determined the *e-ph* coupling constants for Raman active modes of PdTe_2_ according to temperature-dependent Raman spectroscopy [17]. Figure 4a demonstrates the Raman spectrum of the exfoliated PdTe_2_ flake (inset in the panel) with two Raman active modes centered at ~74.4 (*E*_g_, in-plane) and ~132.05 (*A*_1g_, out of plane) cm^−1^, respectively. The *E*_g_ peak exhibits a symmetric lineshape and can be fitted by the Lorentzian/Gaussian function, while the *A*_1g_ peak has an asymmetric lineshape and can be fitted well according to the Breit–Wigner–Fano (BWF) function below [74];(10)$$I\left(\omega \right)=A\;\frac{{\left[q+2\left(\omega -\omega_{i}\right)/\Gamma \right]}^{2}}{1+{\left[2\left(\omega -\omega_{i}\right)/\Gamma \right]}^{2}}$$
where *A* is the Raman scattering intensity, *q* is the asymmetric parameter, and $\omega_{i}$ is the renormalized phonon frequency in the presence of the coupling.

Indeed, the asymmetric lineshape implies a resonance effect resulting from the quantum interference between a discrete state (e.g., here, *A*_1g_ mode) and an electron continuum, suggesting a strong *e-ph* coupling [17,75]. Figure 4b shows the nonlinear redshift of the *E*_g_ frequency shift as a function of temperature dominated by the *4-ph* decay process (blue dashed–dotted line), according to the fitting by Equations (1)–(3), with an ignored thermal expansion coefficient (nearly zero) contribution, as reported previously [76]. Interestingly, the linewidth in Figure 4c shows distinct anomalous behavior with increased temperature, and this behavior cannot fit well according to Equations (4) and (5), in which the rapid decrease at low temperature can be described well according to Equation (9) due to the *e-ph* coupling. Therefore, the *E*_g_ linewidth can be described well using Equations (4), (5) and (9) together, revealing that, at low temperatures, the *e-ph* coupling dominates and rapidly decreases the linewidth broadening. In contrast, with increased temperature, the competition of the *ph-ph* decay processes gives rise to linewidth broadening and is dominated by the *4-ph* contribution above 400 K. Figure 4d displays a nonlinear redshift of *A*_1g_ frequency shift with temperature, dominated by the *3-ph* decay process, as represented by the pink dashed–dotted line, according to the best fitting using Equations (1)–(3), whereas the *4-ph* decay process has little effect above 450 K (blueshift), as indicated by the blue dashed–dotted line. The linewidth of the *A*_1g_ mode in Figure 4e shows moderate nonlinear broadening with increased temperature and can be fitting similarly to the *E*_g_ mode, which results from the *e-ph* coupling contribution up to 150 K; afterward, is dominated by the *ph-ph* decay processes. The asymmetric parameter *q* shows temperature independence (Figure 4f). The *e-ph* constants of the *E*_g_ and *A*_1g_ modes are calculated to be 1.54 and 0.55, respectively, according to Equation (7) in reference [17].

Bi_2_Rh_3_Se_2_ represents a potential platform for exploring *e-ph* coupling and phase transitions, particularly CDW. DFPT calculations at the *Γ* point predicts three acoustic and 39 optical phonon modes at RT, with 18 modes (10 *A*_g_ and 8 *B*_g_) being Raman-active in the 0–300 cm^−1^ range. The RT unpolarized Raman spectrum taken from the as-grown Bi_2_Rh_3_Se_2_ crystal under 532 nm excitation can be divided into two regions [77], region I (~50–100 cm^−1^) and region II (~150–210 cm^−1^), and an isolated peak centered around 233 cm^−1^ ($A_{g}^{10}$), as shown in Figure 5a–d. $A_{g}^{10}$ mode exhibits an asymmetric lineshape, due to the *e-ph* coupling [78], similar to the case of PdTe_2_ [17], and can be fitting well using BWF in Equation (10), as shown in Figure 5b. In addition, there are six Raman-active modes that exhibit asymmetric BWF lineshapes, $A_{g}^{1}$, $A_{g}^{2}$, $A_{g}^{3}$, $A_{g}^{8}$, $A_{g}^{9}$, and $B_{g}^{6}$, located at 60, 68, 75, 168, 178, and 185 cm^−1^, respectively, as shown in Figure 5c,d, while U1, U2, and U3 phonon modes are weak and difficult to identify. Figure 5e represents the second derivative color mapping of the temperature-dependence evolution of Raman active modes for Bi_2_Rh_3_Se_2_, showing two phase transitions around 170 ± 10 K (*T*_1_) and 250 ± 10 K (*T*_2_), as indicated by black dashed lines, similarly to the previously reported phase transition at *T*_s_~240 K [78,79], suggesting the presence of phase symmetry-breaking lower than *T*_2_. With decreased temperature, $A_{g}^{1}$ and $A_{g}^{2}$ peaks in region I merge to one broad peak (C1) centered around 62 cm^−1^ (Figure 5f), while the U1 peak blueshifts. In addition, Raman peaks in region II (Figure 5g) exhibit clear changes, especially below *T*_1_, in which two additional asymmetrical peaks appear, located at 192 (P1) and 207 cm^−1^ (P2), fitting well by BWF in Equation (10), as shown in Figure 5h, implying the presence of a further symmetry-breaking phase at lower temperatures, as well as the *e-ph* coupling. P1 and P2 are assigned to folded phonon modes at the *M*_2_ point due to BZ reconstruction, suggesting the existence of the 2 × 2 CCDW reconstruction below *T*_1_ [79], while, above *T*_1_, these two modes do not appear, in addition to the intensity decreasing, indicating that Bi_2_Rh_3_Se_2_ has the ICCDW phase from *T*_1_ to *T*_2_, similarly to TaSe_2_ and TaS_2_ [80,81]. Figure 5i represents the temperature dependence of the intensity of the CCDW-related P1 and P2 modes, showing nonlinearly behavior rapidly decreased at *T*_1_, which can be described well according to the Landau theory (for second-order phase) [82], taking into account that the interaction of multiple CDWs is disregarded, as the following [83,84];(11)$$X\left(T\right)=X_{0}+A_{i}{\left(1-\frac{T}{T_{CDWi}}\right)}^{2\beta }\;\;\;\;\;\;\;,\;(i=1,2,3,\ldots )$$
where $X_{0}$ refers to the temperature-independent constant, $A_{i}$ is the coupling constant, and β is the critical exponent. According to Equation (11), the extracted parameters of $T_{CDW}$ for P1 and P2 modes are 173 and 170 K, respectively, consistent well to *T*_1_. Similarly, the CCDW-related C1 mode in Figure 5j shows nonlinearly behavior rapidly decreased at *T*_1_, and the C1-split modes related to ICCDW ($A_{g}^{1}$, and $A_{g}^{2}$) are rapidly decreased at *T*_2_, and both can be described well by Equation (11). Figure 5k demonstrates further analysis of the intensity and frequency shift of the CDW-unrelated $A_{g}^{10}$ mode with temperature, showing similar behavior to CDW-related mode changes at *T*_1_ and *T*_2_, indicating there is coupling between electrons and phonons with transition phases CCDW and ICCDW in Bi_2_Rh_3_Se_2_. Moreover, the analysis of temperature dependence of *e-ph* coupling for the $A_{g}^{10}$ and $A_{g}^{3}$ modes reveals that the *e-ph* coupling is different at the CCDW, ICCDW, and the normal phase in Bi_2_Rh_3_Se_2_, suggesting *e-ph* coupling plays an important role in Bi_2_Rh_3_Se_2_ transition phases; for more details, please refer to the Supplemental Material (Figure S10) in reference [77].

### 2.3. Exciton Dynamics-Based PL

Exciton dynamics play an essential role in diverse materials’ PL characteristics, especially in 2D materials, heterostructures, and quantum dots. The activity of excitons, which are bound states of electrons and holes, profoundly impacts the efficiency and properties of PL emissions [85]. Interlayer excitons can be generated in vdW heterostructures, such as those formed by stacking various TMDs through charge transfer or hole tunneling mechanisms [86]. Excitons generally exhibit extended lifetimes and reduced oscillator strengths, resulting in distinctive PL properties, especially at low temperatures, where their presence can significantly influence emission spectra, regardless of their weak intrinsic strength.

For example, the properties of elementary excitations in methylammonium lead iodide (MAPbI_3_), including exciton binding energy and *e-ph* coupling strength, are critical for advancing high-efficiency devices, particularly integrated with TMDs (e.g., MoS_2_), anticipated to enhance performance in optoelectronic device applications [87,88]. Figure 6a shows the AFM image of 4 nm MAPbI_3_ deposited on CVD-grown MoS_2_ (sample II), with denser bright protrusions on c-sapphire (region B), demonstrating nucleation-limited growth, while the triangular region of MoS_2_ (region A) appears as atomically smooth with no changes except for some features on its edges (Frank–van der Merwe mode) [89]. The varying growth behaviors of MAPbI_3_ on two substrates can be ascribed to the differences in hydrophilicity and reduced nucleation energy on MoS_2_ [89,90]. Figure 6b represents the AFM image of 8 nm MAPbI_3_ deposition on MoS_2_ (sample III), with a roughness surface of MoS_2_ (~60 nm), covered by clusters, suggesting a change in the growth mode of MAPbI_3_ on MoS_2_ to Stranski–Krastanov mode.

To better understand the exciton dynamics, Figure 6c shows the evolution of PL spectra with temperature for sample III in Figure 6b, revealing a dominant emission peak center at 1.64 eV (peak I), which is assigned to the tetragonal-I phase of MAPbI_3_ [91], while below 150 K a new emission peak centered at 1.69 eV (peak II) belongs to the orthorhombic phase. These observed emission peaks can be attributed to the recombination of free excitons [89,92]. At 100 K, a new emerging emission peak (1.56 eV, peak III) at the low-energy side appears, which is ascribed to trap-mediated exciton radiative recombination [93]. Figure 5d displays the extracted photon energy as a function of temperature for the three peaks, showing that peak I redshifts from 300 to 150 K, similar to pure MAPbI_3_ behavior [91], while from 150 to 100 K it exhibits blueshift, indicating that transition from tetragonal to orthorhombic starts to occur at 150 K. Peak II in the orthorhombic phase keeps the trend to redshift. Figure 6e displays the relative percentage evolution as a function of the temperature for the three emission peaks. As the temperature diminishes, the proportion of the tetragonal phase consistently declines, whereas that of the orthorhombic phase increases. The percentage of the trapped exciton peak (peak III) increases at low temperatures, which is ascribed to the lower binding energy of trapped excitons and transitions of surface states.

Different from sample III, the evolution of PL spectra taken from region A in sample II (Figure 6a) shows the main emission peak centered at 1.68 eV (IV), which is increased as the temperature decreases from 300 to 120 K (Figure 6f), attributed to the enhanced optical matrix elements due to strengthened Coulomb attraction and reflects radiative recombination of free-exciton behavior [92]. The absence of a trapped exciton peak indicates a lower defect density in sample II. Figure 6g displays the photon energy of sample II as a function of the temperature. Different from sample III, there is a turning point at 260 K (peak IV), which exhibits redshift with decreased temperature from RT to 260 K, then turns to blueshift until 120 K; below 120 K, a new peak appears at 1.72 eV, assigned to orthorhombic phase MAPbI_3_. In addition, the photon energy of MAPbI_3_/sapphire (region B) shows a similar trend (black dots) except at the temperature range 300–260 K. This behavior of photon energy in sample II suggests MAPbI_3_ in the cubic phase formed at RT on MoS_2_, and then the second-order phase transition occurs from cubic to another different tetragonal phase at 260 K, named tetragonal-II. Additionally, as the temperature drops below 120 K, a transition from tetragonal-II to the orthorhombic occurs. In the case of MAPbI_3_/sapphire, a phase transition from tetragonal-II to orthorhombic occurs at 100 K. Figure 6h represents the linewidth of peak IV with temperature showing linear broadening and can be described well using the one-oscillator model of the *e-ph* interaction; more details in the reference [89].

Figure 7a demonstrates the AFM image of 3 nm PbI_2_ deposited on MoSe_2_ with roughness features [94], indicating its vdW epitaxial growth is not optimal like that on MoS_2_ at RT [95]. The bottom-inserted magnified AFM image with its profile reveals the emergence of single-layered islands with smooth surfaces on the incomplete PbI_2_ layers, suggesting a quasi-layer-by-layer vdW epitaxial growth mode of PbI_2_ on MoSe_2_ at RT. The Raman spectrum insert in the upper panel taken from PbI_2_ on MoSe_2_ shows about five distinct peaks, two peaks assigned to *A*_1g_ (339.6 cm^−1^) and *E*_g_^1^ (289.1 cm^−1^) modes for MoSe_2_ and three peaks assigned to *E*_g_ (70.1 cm^−1^), *A*_1g_ (94.1 cm^−1^), and *A*_2u_ (108.1 cm^−1^) related to PbI_2_. Before deposited PbI_2_, the phonon modes of MoSe_2_ are located at 239.4 (*A*_1g_) and 289. 5 cm^−1^ (*E*_g_^1^), respectively, while after deposited PbI_2_ on MoSe_2_, *A*_1g_ blueshifts by 0.2 and *E*_g_^1^ redshifts by 0.4. Such shifts are ascribed to the interlayer coupling after deposited PbI_2_, the *e-ph* coupling effect at the interface, or interfacial charge transfer from MoSe_2_ to PbI_2_ [94,96]. The upper panel in Figure 7b shows the PL spectra of exfoliated monolayer MoSe_2_ with two fitted emission peaks, $A^{0}$ (1.57) and $A^{-}$ (1.52 eV), whereas the bottom panel displays the PL spectra of PbI_2_ deposited on MoSe_2_. The intensity of the peak emission quenches due to interface band alignment, with an additional peak appearing at 1.43 eV, arising from the interlayer excitons [95]. Figure 7c illustrates the temperature-dependent evolution of PL spectra for the PbI_2_/MoSe_2_ vdW heterostructure from 90 K to 300 K. At a low temperature, a new peak at 1.51 eV appears, indicating it is related to a defect-bound exciton [97], which disappears at 150 K and above. Figure 7d represents the peak positions with temperature, indicating that intrinsic three peaks redshift as the temperature increases, attributed to *e-ph* interaction and the slight bonding length enlargement, similar to the case of PbI_2_/MoS_2_ [95]; for more details, please refer to the reference [94].

### 2.4. Phonon Anisotropy

Layered materials with a puckered honeycomb configuration have significant in-plane anisotropy, making them ideal platforms for advanced technologies. Among them, Black Arsenic (b-As) exhibits ambient stability [98]. Figure 8a represents the typical AFM image of exfoliated b-As nanoflakes of different thicknesses, 11 (region I), 40 (region II), and 58 nm (region III) [99]. The b-As nanoflake is spin-coated by a polymethyl methacrylate (PMMA) thin film after AFM measurement to protect it from the laser effect of Raman spectroscopy. Figure 8b displays the unpolarized Raman spectra of b-As nanoflakes with/without coating of PMMA, in which both show similar Raman active modes, centered around 221 ($A_{g}^{1}$), 227 (*B*_2g_), and 255 cm^−1^ ($A_{g}^{2}$), respectively. Figure 8c–h represent the color mappings of angle-resolved polarized Raman spectroscopy (ARPR) of b-As nanoflakes at different thicknesses (marked by the circles in panel a) in a parallel-polarization configuration (PPC) and in different excitations (532 and 633 nm). All Raman active modes exhibit periodic variation with θ. $A_{g}^{1}$ and $A_{g}^{2}$ show a periodic of 180°, whereas *B*_2g_ exhibits a periodic of 90°, indicating that the polarized Raman intensity has a robust relation to the crystalline orientation of b-As [99]. Interestingly, the ARPR of the $A_{g}^{1}$ and $A_{g}^{2}$ modes significantly varies with the flake thickness and excitation.

The corresponding polar plots of Raman intensities for both modes are shown in Figure 8i–w. The polarized Raman intensities can be described according to the classical Placzek approximation below [100];(12)$$I\propto {\left|e_{s}Re_{i}^{\tau }\right|}^{2}$$
where *R* is the Raman tensor, and $e_{i}$ and $e_{s}$ are the incident and scattered vectors of the laser. In the PPC, $e_{i}$ = $e_{s}$= (cos⁡θ,0,sin⁡θ). The Raman intensities according to the real Raman tensors are too simple to describe the $A_{g}^{1}$ and $A_{g}^{2}$ modes well; therefore, it is better to consider the influence of light absorption and the Raman tensor elements should have complex values; for more details, please refer to the reference [99]. Accordingly, the intensities are as follows;(13)$$I_{A_{g}}\propto {\left|c\right|}^{2}\sin^{4}\theta +{\left|a\right|}^{2}\cos^{4}\theta +2\left|ac\right|\cos^{2}\theta \;{sin}^{2}\theta \;cos\varphi_{ac}$$(14)$$I_{B_{2g}}\propto {\left|d\right|}^{2}\sin^{2}2\theta$$

Raman intensities in the polar plots in Figure 8i–w are described well by Equations (13) and (14) (red solid lines), in which the *B*_2g_ mode exhibits the same polarization independent of the excitation wavelength and b-As flake thickness, producing four-lobed patterns. In contrast, $A_{g}^{1}$ and $A_{g}^{2}$ modes exhibit polarization dependent on the excitation wavelength and flake thickness, with a bow tie-like shape intensity maximum at 0° or 90. The black solid lines represent the fitting after correction by considering the interference and birefringence effects, revealing that the $A_{g}^{1}$ mode displays similar anisotropy, regardless of given conditions, while $A_{g}^{2}$ exhibits anisotropy dependent on the excitation wavelength. This anisotropy of phonons is useful for determining the crystal orientation of b-As.

As explained in the discussion above, Bi_2_Rh_3_Se_2_ crystal exhibits two transition phases related to CDW, with different *e-ph* coupling in both transition phases; therefore, investigating in-plane phonon and *e-ph* coupling anisotropies is crucial. In this context, the ARPR spectra of Bi_2_Rh_3_Se_2_ crystal upon laser excitations 532 and 633 nm in the PPC are shown in Figure 9a,b, represented by 3D color maps. $A_{g}^{1}$, $A_{g}^{2}$, $A_{g}^{3}$, $A_{g}^{8}$, $A_{g}^{9}$, $A_{g}^{10}$ and $B_{g}^{6}$ mode intensities (*I*) exhibit periodic variation with *θ*, signifying in-plane phonon anisotropy [77]. Among them, $A_{g}^{3}$, $A_{g}^{8}$, $A_{g}^{10}$ and $B_{g}^{6}$ modes show polarization independence on the excitation wavelength. Taking $A_{g}^{10}\;$as an example, the polar plots of *I* in Figure 9c,d show a bow tie-like shape with an intensity maximum ~127° (307°) under both excitation wavelengths 532 nm (green dots) and 633 nm (red dots), fitting well according to Placzek approximation Equation (12). The extracted 1/q of the $A_{g}^{10}$ mode under excitation wavelengths (532 and 633 nm) shown in Figure 9e,f with a bow tie-like shape varies with *θ*, indicating the *e-ph* coupling anisotropy, in which the 1/q maximum intensity is along the *b*-axis, while the maximum intensity of $A_{g}^{10}$ is along the *a*-axis, suggesting that *e-ph* coupling appears to suppress *I* in Bi_2_Rh_3_Se_2_ [75]. It is important to mention that the absolute value of 1/q upon excitation under 633 nm (~0.2) is around twice that under 532 nm (~0.1), indicating a strong *e-ph* coupling under excitation wavelength 633 nm. Figure 9g,h represent the anisotropic ratios (the absolute value/the absolute minimum value) of *I* and 1/q with *θ*, revealing that both ratios under the excitation wavelength of 633 nm are larger than those under 532 nm, implying strong in-plane anisotropy under excitation wavelength 633 nm.

Twisting between layered materials frequently results in twist angle-dependent unique physical characteristics. The twisting bi-layer (tBL) and twisting tri-layer (tTL) α-MoO_3_ homojunctions in different *φ* values can be formed using the broken α-MoO_3_ pieces, as shown in Figure 10a, in which the *φ* of tBL α-MoO_3_ is defined as the angle between the direction [001] of the bottom and top nanoribbons (shown by scheme inset in the panel). Figure 10b represents the Raman spectrum of a α-MoO_3_ nanoribbon (red) with a high-frequency band (SVPM, 818 cm^−1^) and low-frequency band (translational vibration phonon mode TVPM, 156 cm^−1^) as explained in Section 2.1 above, and compared with the Raman spectrum of SiO_2_/Si substrate (black). ARPR scattering measurements were conducted on SL, tBL, and tTL α-MoO_3_ upon 532 and 633 nm excitation in the PPC to assess the layer contributions to *I* in the twisted α-MoO_3_ homojunctions, in which all phonon modes periodically change with the polarized angle upon both laser excitations, indicating phonon anisotropy; more details in reference [101]. For example, Figure 10c–j show the polar plots of *I*_SVPM_ at different *φ* for tBL α-MoO_3_ (black dots) and its top (blue dots) and bottom (red dots) layer upon the laser excitations 532 and 633 nm. The polar plots of the top and bottom layer show a two-folded shape with *I*_SVPM_ maximum along the direction of SL α-MoO_3_ [100], while the two-folded shape of the *I*_TVPM_ maximum direction shows a rotation of 90° compared with the *I*_SVPM_ maximum direction, as shown in Figure 10k–r. Significantly, the *I*_SVPM_ from SL is analogous to that from tBL α-MoO_3_ upon 532 nm excitation; however, *I*_SVPM_ from SL α-MoO_3_ is much lower than that from tBL α-MoO_3_ upon 633 nm excitation, indicating the layer contribution to *I*_SVPM_ of tBL α-MoO_3_ under 633 nm excitation more than that under 532 nm, similarly to *I*_TVPM_. Moreover, the symmetry of SVPM and TVPM changes from a two-fold pattern to a propeller-shaped pattern with increased *φ*, similar to those reported with ReS_2_ and ReSe_2_ [102,103]. Therefore, it is essential to look into the contributions from each layer individually to assess the evolution of *I* with the polarization angle, as described according to Placzek approximation and the intensities in the equations in the reference [101], represented by the curves in Figure 10c–r: blue curve (top layer), red curve (bottom layer), and the contribution of tBL (black curve). According to the best fitting (black curve), the extracted parameters *m*_532_ (top layer) and *n*_532_ (bottom layer) are close to 0.65, suggesting the top and bottom layers contribute equally to *I*_SVPM_ in tBL α-MoO_3_ under 532 nm excitation, while *m*_633_ > 1.7 and *n*_633_~1.1, indicating the top layer contributes to *I*_SVPM_ more than that bottom layer under 633 nm excitation. Considering the layer-dependent interference effect, the recalculated enhancement and contribution factors are part of the same interval [101], although they do not match each other well. Therefore, the intensity (*I*) anomaly cannot be ascribed only to the layer-dependent interference effect; more physical mechanisms need additional exploration in the future.

vdW heterostructures composed of TMDs in 2D/2D or 2D/3D configurations have exceptional characteristics for advanced electronics, tunneling transistors, and catalytic applications, for instance, high-quality MoS_2_/MoO_2_ heterostructures prepared by one-step chemical vapor deposition (CVD) [104]. Figure 11a demonstrates the OM image of MoS_2_ (purple regular triangle) on MoO_2_ in high quality, as confirmed by the PL (1.864 eV) and Raman (404 cm^−1^) mappings inserted in the corners of the panel. Figure 11b shows the Raman spectra of MoO_2_, MoS_2_, and MoS_2_/MoO_2_ (with additional peaks centered at 384 and 404cm^−1^ for SL MoS_2_), confirming the heterostructure. Indeed, the higher work function of MoO_2_ (5.5–5.7 eV) [105] in the MoS_2_/MoO_2_ heterostructure causes interfacial charge transfer from MoS_2_ to MoO_2_, thus altering the optical characteristics of MoS_2_ inside these heterostructures, similarly to those MoS_2_/CrOCl heterostructures reported previously [106]. This could be confirmed by investigating the underlying monoclinic MoO_2_ on the epitaxial MoS_2_ via ARPR and ARPPL measurements in the PPC. Figure 11c displays the color map of ARPR for MoS_2_/MoO_2_ heterostructures, revealing periodic variation properties of MoO_2_ phonon modes, e.g., *A*_g_ mode (125 cm^−1^) with a two-fold symmetry feature, as shown by the intensity maximum along the MoO_2_ [201] direction in the polar plot (Figure 11d), similarly to the previous report [107]. The magnified color map in Figure 11e represents the ARPR of the *A*_1g_ mode (404 cm^−1^) for MoS_2_ in the MoS_2_/MoO_2_ heterostructure with two-fold symmetry. The corresponding polar plot in Figure 11f also displays a two-fold symmetry for *A*_1g_ (red) with the intensity maximum along MoO_2_ [201] direction, different from those of monolayer MoS_2_ on c-sapphire (blue circle, isotropic property). The ARPPL of MoS_2_ on MoO_2_ exhibits identical two-fold symmetry, as shown in Figure 11g,h. The anisotropy ratios are calculated for Raman and PL of MoS_2_/MoO_2_ to be 1.27 and 1.29, respectively; more details in reference [104]. The in-plane anisotropy in MoS_2_ is mostly ascribed to anisotropic interfacial charge interactions in MoS_2_/MoO_2_ heterostructure induced by MoO_2_; such behaviors have been reported for CsPbBr_3_ and CuPc on ReS_2_ [108,109].

Twisting isotropic–anisotropic vdW heterostructures offer a means to control 2D materials’ electronic and phononic characteristics and induce in-plane anisotropy in certain isotropic materials. Fabricated MoS_2_/CrOCl heterostructures with different twisting angles are shown in Figure 12a [110]. The nanoflakes of exfoliated CrOCl display a stripe pattern, with the long axis aligned with the [010] CrOCl direction, in which Young’s modulus is significantly greater along this direction compared with the [100] CrOCl direction. A fascinating property of MoS_2_/CrOCl heterostructures is the ability to control the in-plane anisotropy of phonons and excitons through twisting angles (φ= 0°, 7°, 25°, 31°, 43°, 46°, and 53°). The extracted fitted ratios of $E_{g}^{2}$ and $A_{g}^{1}$ modes of twisted MoS_2_/CrOCl heterostructures are presented in Figure 12b, revealing periodic changes with the twist angle φ, fitting well by a cos^2^φ. The larger anisotropy ratios for both phonon modes ($E_{g}^{2}=$1.22, and $A_{g}^{1}=$1.15) at the twisting angle 0° and the smallest values ($E_{g}^{2}=$1.13, and $A_{g}^{1}=$1.09) at the twisting angle 31° suggest that varying twisting angles permit a continuous tuning of the generated in-plane anisotropy across a large range. A similar trend is observed for the anisotropy ratios of the polarized PL intensities, as shown in Figure 12c. The dependency on the twist angle arises from the anisotropic carrier mobility created by the localized charge distribution of the anisotropic CrOCl substrates, which is further modulated by the uniaxial local confinement of charge carriers resulting from the 1D moiré pattern, according to the first principles calculation results in reference [110].

## 3. Conclusions

Many-body interaction effects, including *e-ph* coupling, *ph-ph* interactions, and the excitons dynamics, etc., play an important role in material properties and behaviors, and can control the superconductivity, thermal conductivity, and optical responses of 2D materials, eventually deepening our knowledge of condensed matter physics and enabling advancements in technology and materials science. Temperature-dependent Raman spectroscopy (TDRS) and adopted theoretical models provide a reliable insight into modes’ thermal behaviors with temperature. For instance, the G mode of 7-AGNRs linearly redshifts with temperature, whereas the frequency shift of RBLM is nonlinearly redshifted and is mainly dominated by the *4-ph* decay process. The linewidths are nonlinearly broadening, driven by the competition between the *3-* and *4-ph* decay processes. These thermal properties of 7-AGNRs will be useful for application-based AGNRs in optical and electrical devices in the future. In addition, TDRS analysis demonstrated the substrate effect on graphene (different layers) rather than the anharmonic *ph-ph* interaction or thermal expansion. The *ph-ph* scattering processes of representative SVPMs of layered α-MoO_3_ are primarily dominated by giant *4-ph* decay, indicating its potential application in thermal devices as an emerging material with ultra-low phonon thermal conductivity and high carrier mobility. TDRS of PdTe_2_ revealed a strong *e-ph* coupling based on the linewidths broadening analysis with temperature, confirming it is a phonon-mediated BCS-type superconductor, which is crucial for quantum computing applications and information processing. Furthermore, TDRS provides deep insight into the phase transitions of materials with temperature. For instance, Bi_2_Rh_3_Se_2_ exhibited two phases with different *e-ph* couplings: the 2 × 2 CCDW phase (below~170 K), with zone-folding phonon modes P1, P2, and the ICCDW phase (~170 to ~250 K). These results indicate that Bi_2_Rh_3_Se_2_ may serve as a suitable material for examining the interaction between CDW and superconductivity in future work, particularly through the variations in *e-ph* coupling associated with phase transitions. Different emission properties of the heterostructures (MAPbI_3_/MoS_2_ and PbI_2_/MoSe_2_) as a result of interfacial excitons were revealed by temperature-dependent PL measurements, which opens the door to the possibility of building molecularly thin heterostructures and offers a platform for studying innovative applications. The ARPR and ARPPL results showed valuable insights into the anomalous phonon polarization responses in low symmetric layered 2D materials, potentially facilitating the development of future technologies based on tunable phonon polarization, as well as engineering and controllably regulating induced in-plane optical anisotropy in isotropic/anisotropic heterostructures.

Finally, investigating the interactions of quasi-particles in 2D vdW materials via Raman spectroscopy and PL is essential for understanding these materials’ electronic, optical, and thermal characteristics. Continued progress in Raman spectroscopy methods, especially when combined with external magnetic, electric, and strain fields, will further expand our comprehension of these intriguing materials and broaden the potential applications for future research.

## Figures and Tables

**Figure 2 nanomaterials-15-00101-f002:**
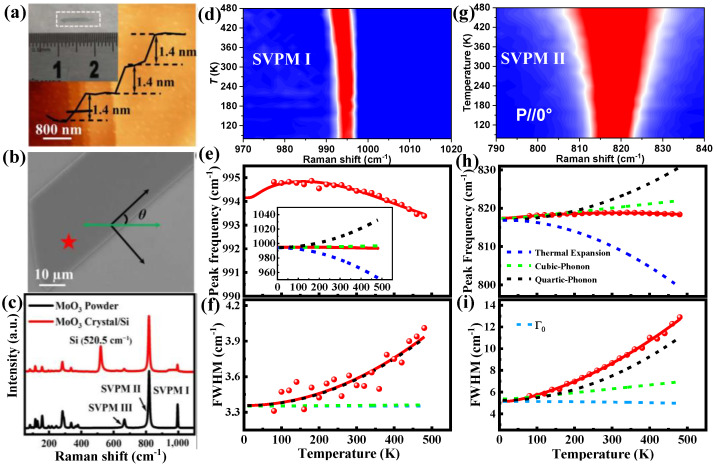
(**a**) Typical AFM image of layered α-MoO_3_ flake edges, with a height profile step of ~1.4 nm. Inset: an OM image in centimeter size. (**b**) SEM image of the α-MoO_3_ flake measured by Raman spectroscopy. (**c**) Raman spectra at RT for MoO_3_ powders and as-grown α-MoO_3_ flakes. (**d**,**g**) Color mapping evolution of temperature polarized dependence of SVPMs I and II for layered α-MoO_3_ flake. (**e**,**f**,**h**,**i**) Extracted Raman frequency shift and linewidth as a function of temperature of SVPMs I and II, fitted by Equations (3)–(5), respectively. The solid red line represents the best-fitting of data. The green, black, blue, and light blue dashed lines represent the contribution of *3-ph* and *4-ph* decay processes, thermal expansion, and *Γ*_0_, respectively. Reproduced with permission from [59], copyright 2022, Springer Nature.

**Figure 3 nanomaterials-15-00101-f003:**
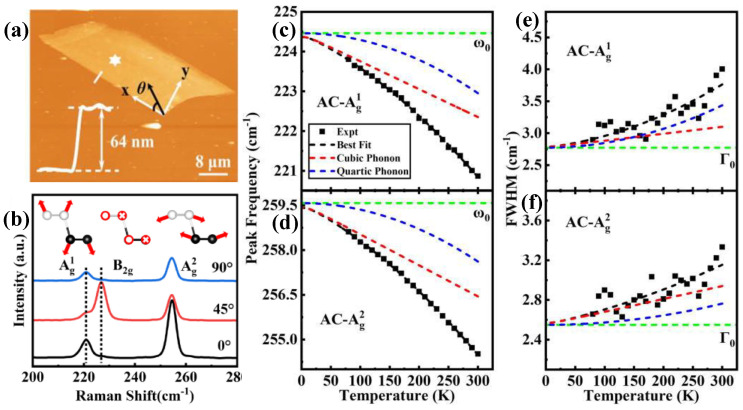
(**a**) AFM image of b-As flake of 64 nm thickness, represented by height profile. The polarization angle (θ) is defined to be 0° when the incident polarization is along the *x*-axis. (**b**) Measured polarization Raman spectra of b-As flake in RT and at different incident angles of 0, 45, 90°. Inset: the atoms’ vibrations displacement. (**c**–**f**) Peak frequency and linewidth of A^1^_g_ and A^2^_g_ phonon modes in AC direction, fitted by Equations (3)–(5), respectively. The black dashed line represents the best fitting of data. The red and blue dashed lines are the *3-* and *4-ph* contributions, respectively. Reproduced with permission from [61], copyright 2022, AIP Publishing.

**Figure 4 nanomaterials-15-00101-f004:**
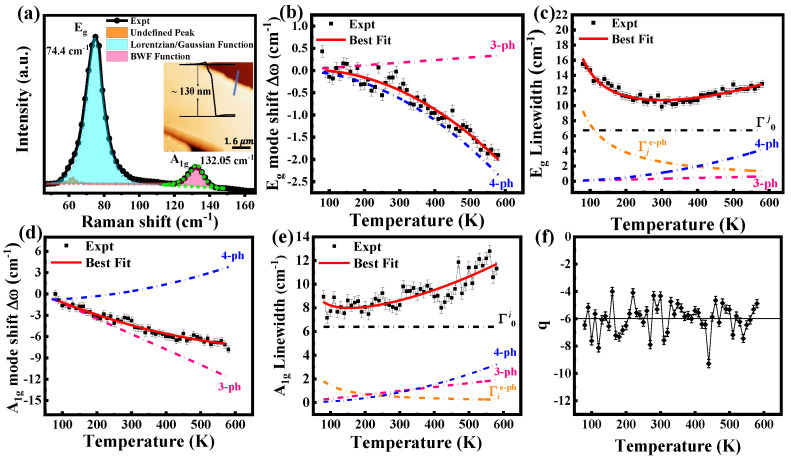
(**a**) Raman spectrum of exfoliated PdTe_2_ nanoflake of thickness 130 nm represented by inserted AFM image. *E*_g_ and *A*_1g_ peaks fitted by Lorentzian/Gaussian and BWF functions, respectively. (**b**–**e**) Raman frequency shift and linewidth as a function of the temperature of *E*_g_ and *A*_1g_ phonon modes for exfoliated PdTe_2_ nanoflake, respectively. The solid red lines are the best-fitting data according to Equations (3), (4), (5) and (9). The yellow, pink, blue, and black dashed–dotted lines represent the contributions of *e-ph* coupling, *3-* and *4-ph* interactions, and *Γ*_0_, respectively. (**f**) BWF asymmetric factor *q* at different temperatures. Reproduced with permission from [17], copyright 2022, John Wiley & Sons.

**Figure 5 nanomaterials-15-00101-f005:**
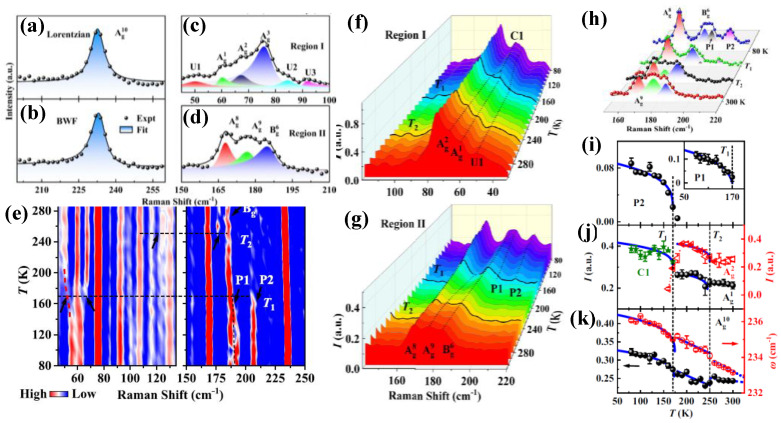
(**a**,**b**) represent the *A*_g_^10^ phonon mode fitted by Lorentzian and BWF functions, respectively. (**c**,**d**) Regions I and II of Raman active modes of as-grown Bi_2_Rh_3_Se_2_ crystal under 532 nm excitation at RT. (**e**) Color mapping evolution of temperature dependence for as-grown Bi_2_Rh_3_Se_2_. (**f**,**g**) Zoomed-in Raman spectra for regions I and II, respectively. (**h**) Selected peaks that are fitted well by BWF function at 80 K, T_1_, T_2_, and RT. Intensity (*I*) evolution with temperature of P2 (inset P1) (**i**), C1, *A*_g_^1^, and *A*_g_^2^ modes (**j**). (**k**) Temperature dependence of *I* and ω for *A*_g_^10^ mode. Both *I* and ω of these modes are fitted well by Equation (11). Reproduced with permission from [77], copyright 2023, American Physical Society.

**Figure 6 nanomaterials-15-00101-f006:**
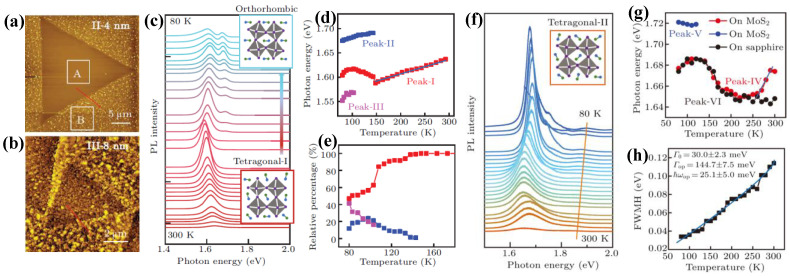
AFM images of co-deposited MAPbI_3_ ultrathin films/MoS_2_ in different thicknesses 4 nm sample II (**a**), and 8 nm sample III (**b**). (**c**) Evolution of PL spectra with temperature (80 K-RT) taken from sample III in panel b. Inset: atomic models of orthorhombic and tetragonal-I phases. (**d**) The corresponding extracted data of photon energy for peaks I, II, and III, respectively, as a function of temperature (red color represents a tetragonal phase, blue is an orthorhombic phase, and purple is the defect peak). (**e**) Proportional fraction of three peaks in the MAPbI_3_/MoS_2_ heterostructure. (**f**) Evolution of PL spectra taken from sample II in panel a. Inset: atomic model of tetragonal-II phase. Extracted data of (**g**) photon energy (red color is a tetragonal phase, and blue is an orthorhombic phase) and pure MAPbI_3_ (black), and (**h**) linewidth of tetragonal phase as a function of temperature, respectively. Reproduced with permission from [89], copyright 2023, Chinese Physics B.

**Figure 7 nanomaterials-15-00101-f007:**
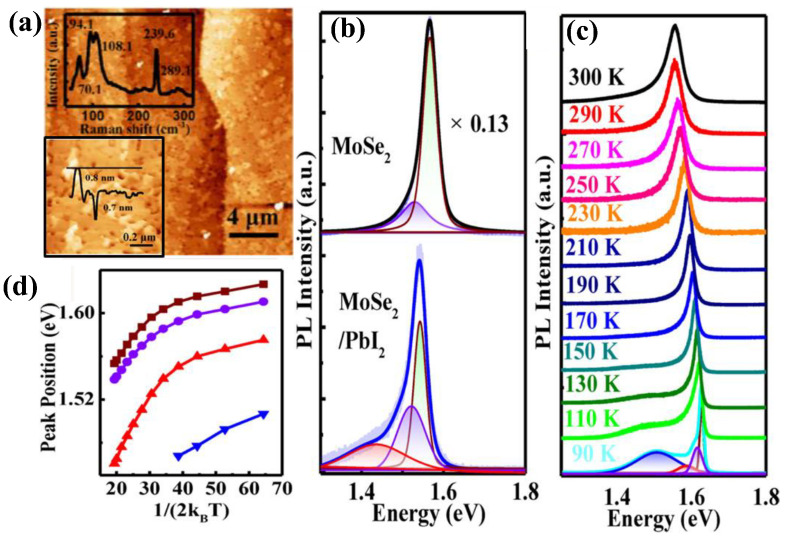
(**a**) AFM image of PbI_2_/MoSe_2_ vdW heterostructure. Inset: the Raman spectrum (**upper**), and 3D AFM image (**bottom**) with inserted profile indicating a quasi-layer-by-layer growth mode. (**b**) PL of MoSe_2_ (**upper**) and PbI_2_/MoSe_2_ (**bottom**). (**c**) Evolution of temperature dependence of PL of vdW heterostructure PbI_2_/MoSe_2_. (**d**) PL peak position as a function of temperature. Reproduced with permission from [94], copyright 2020, American Chemical Society.

**Figure 8 nanomaterials-15-00101-f008:**
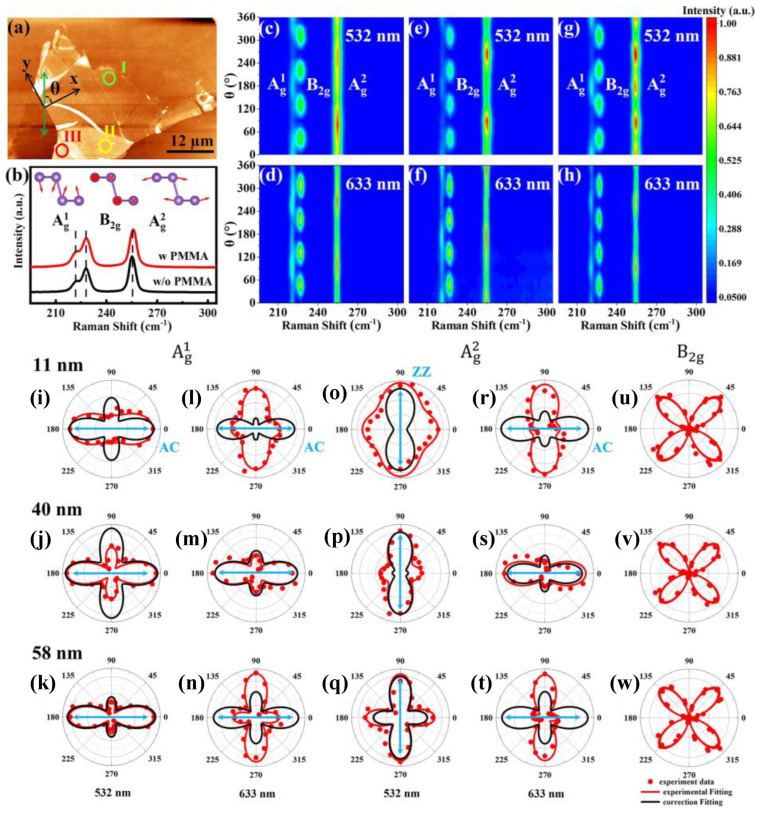
(**a**) Typical AFM image of as-exfoliated b-As flakes of different thicknesses, with inserted circles as green, yellow, and red, indicating regions I (11 nm), II (40 nm), and III (58 nm), respectively. θ is the defined angle between the *x*-axis (flake) and incident polarization. (**b**) Raman spectra of b-As flakes without and with PMMA-protected layer. Inset: the atomic vibration displacement of *A*_g_^1^, *B*_g_^2^, and *A*_g_^2^ modes. The color mapping of polarized Raman spectroscopy using laser excitation 532 and 633 nm in PPC for regions I (**c**,**d**), II (**e**,**f**), and III (**g**,**h**), respectively. (**i**–**w**) Polar plots of *A*_g_^1^, *A*_g_^2^, and *B*_g_^2^ phonon modes for as-exfoliated b-As flakes. The red and black solid lines represent the best fitting without (Equations (13) and (14)) and with correction, respectively. Reproduced with permission from [99], copyright 2021, American Chemical Society.

**Figure 9 nanomaterials-15-00101-f009:**
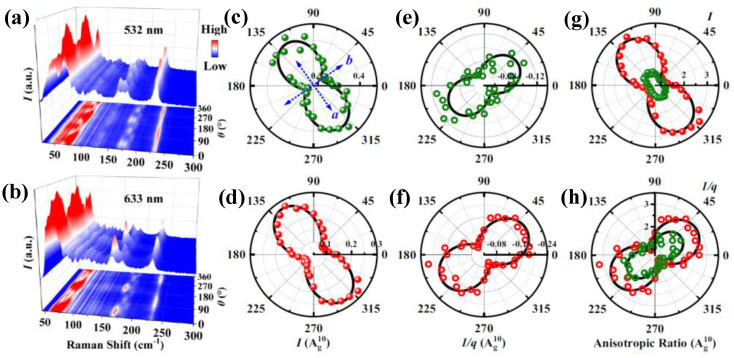
(**a**,**b**) 3D color map with the projection of ARPRS of Bi_2_Rh_3_Se_2_ excited by laser 532 and 633 nm in PPC. Polar plots dependence of *I* (**c**,**d**), asymmetric factor 1/q (**e**,**f**), and their anisotropic ratios (**g**,**h**) for *A*_g_^10^ phonon mode. The green and red dots are the extracted data under laser excitation 532 and 633 nm. The solid lines represent the best-fitting data. Reproduced with permission from [77], copyright 2023, American Physical Society.

**Figure 10 nanomaterials-15-00101-f010:**
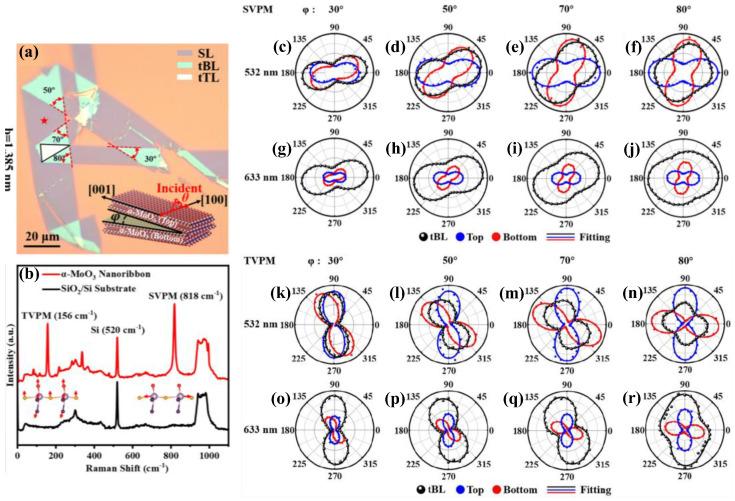
(**a**) OM image of twisting α-MoO_3_. Insert: the tBL α-MoO_3_ scheme with a twisting angle (φ). (**b**) Raman spectra of α-MoO_3_ and SiO_2_/Si substrate. Inset: atomic vibration displacements of TVPM and SVPM. (**c**–**r**) Polar plots of SVPM and TVPM for tBL α-MoO_3_ in different φ, under laser excitations 532 and 633 nm, respectively. The solid lines represent the best fitting experimental data. Reproduced with permission from [101], copyright 2024, American Chemical Society.

**Figure 11 nanomaterials-15-00101-f011:**
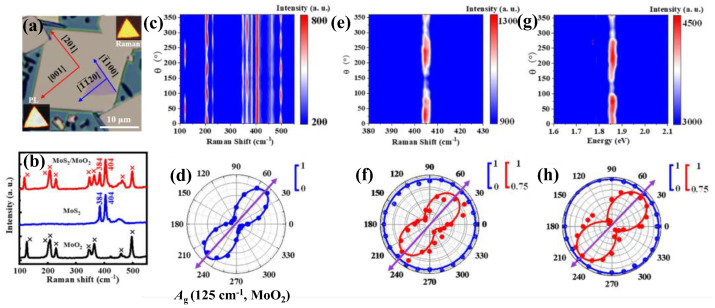
(**a**) OM image of single triangular MoS_2_ on MoO_2_. Inset: in the lower-left corner is the PL mapping, and in the upper-right corner is Raman mapping (404 cm^−1^). (**b**) Raman spectra of MoO_2_ (black), MoS_2_ (blue), and CVD-grown MoS_2_/MoO_2_ heterostructure (red). (**c**) Color mapping of ARPR for MoS_2_/MoO_2_ in the PPC. (**d**) Corresponding polar plot of *A*_g_ mode of MoO_2_ (125 cm^−1^). (**e**) Magnified color mapping of *A*_1g_ mode of MoS_2_ (404 cm^−1^). (**f**) The corresponding polar plot of MoS_2_/c-sapphire (blue), and MoS_2_/MoO_2_ (red). (**g**) Color mapping intensity of polarized PL of MoS_2_/MoO_2_. (**h**) The corresponding polar plot of MoS_2_/c-sapphire and MoS_2_/MoO_2_. The fitting data are represented by solid lines. Reproduced with permission from [104], copyright 2023, AIP Publishing.

**Figure 12 nanomaterials-15-00101-f012:**
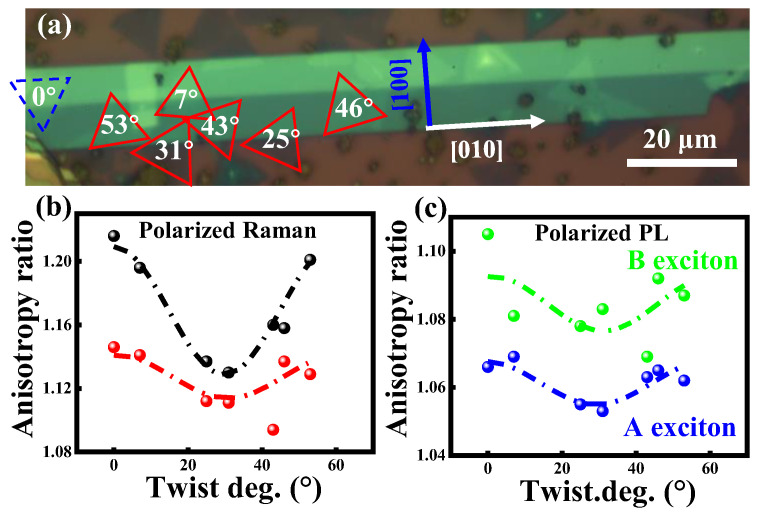
(**a**) OM image of MoS_2_/CrOCl heterostructure. The monolayers of MoS_2_ in different twisting angles with CrOCl crystal direction are marked. Twisting angles’ dependent anisotropy ratios of polarized Raman (**b**) and PL exciton intensities (**c**). Reproduced with permission from [110], copyright 2024, AIP Publishing.

**Table 1 nanomaterials-15-00101-t001:** Raman bands of materials addressed in this review.

Materials	Raman Modes	$\mathrm{Positions}\;\omega_{0}$ (cm^−1^)	Origin	Figure
7-AGNRs	G	1595	in-plane C-C bond stretching vibrations	(1)
	CH_1_	1220	the edge C-H bond bending vibrations	
	CH_2_	1255		
	D	1340	the edges breaking the periodicity of a perfect honeycomb lattice	
	RBLM	396	the collective vibrations along the width direction of 7-AGNRs	
MoO_3_	SVPM I	995	the stretching vibrations of Mo-O_1_ along [010]	(2)
	SVPM II	819	the stretching vibrations of Mo-O_2_ along [010]	
	SVPM III	666	the stretching vibrations of Mo-O_3_ along [100]	
	TVPM	156	the rigid MoO_4_ translational vibrations	(10)
b-As	$A_{g}^{1}$	221	the coupled in-plane and out-of-plane vibrations mainly along [001] and [010]	(3, 8)
	$A_{g}^{2}$	255		
	*B* _2g_	227	in-plane vibrations along [100]	
PdTe_2_	*E* _g_	74	in-plane vibrations of Te atoms in opposite directions	(4)
	*A* _1g_	132	out-of-plane vibrations of Te atoms in opposite directions	
Bi_2_Rh_3_Se_2_	$A_{g}^{10}$	233	most intense vibration but not assigned	(5, 9)
	P1	192	zone-folding modes at the M_2_ point due to charge density wave states	
	P2	207		
MoS_2_/MoO_2_	*A* _g_	125	the bending vibrations of Mo-Mo bond	(11)
	*A* _1g_	404	out-of-plane vibrations of S atoms of MoS_2_ in opposite directions	

## Data Availability

No new data were created or analyzed in this study.

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
