# Peer review of "Raman and Photoluminescence Studies of Quasiparticles in van der Waals Materials"

_nanomaterials, 2025, doi:10.3390/nano15020101_

Round 1

Reviewer 1 Report

Comments and Suggestions for Authors
  • While the review presents valuable information, certain parts of the text are difficult to read. A suggestion to enhance readability and clarity is to prepare systematic tables summarizing the e.g. Raman bands discussed starting from Chapter 2. This would provide a concise reference point for the reader and facilitate a quicker understanding of key findings, trends, and assignments of the Raman spectra.
  • The included figures, while informative, are very small, making it challenging to get details. In several cases, the attempt to save space appears to compromise the figure quality and effectiveness. Larger images will improve visibility and understanding.  Ensure all axes are labeled properly with clear font, units, and descriptions for better comprehension.  Use adequately sized fonts and high-resolution images to make the content legible. Review for potential typos in figure captions or labels.
  • The conclusion section falls short of adequately summarizing the main findings of the review. It would be beneficial to clearly outline the key advancements in the field discussed in the review. Highlight significant gaps in knowledge or areas requiring further research. Emphasize the implications of the findings for future work or practical applications. Provide a concise, insightful synthesis that ties together the central themes of the review.

Author Response

Reviewer #1: While the review presents valuable information, certain parts of the text are difficult to read. A suggestion to enhance readability and clarity is to prepare systematic tables summarizing the e.g. Raman bands discussed starting from Chapter 2. This would provide a concise reference point for the reader and facilitate a quicker understanding of key findings, trends, and assignments of the Raman spectra. The included figures, while informative, are very small, making it challenging to get details. In several cases, the attempt to save space appears to compromise the figure quality and effectiveness. Larger images will improve visibility and understanding.  Ensure all axes are labeled properly with clear font, units, and descriptions for better comprehension.  Use adequately sized fonts and high-resolution images to make the content legible. Review for potential typos in figure captions or labels. The conclusion section falls short of adequately summarizing the main findings of the review. It would be beneficial to clearly outline the key advancements in the field discussed in the review. Highlight significant gaps in knowledge or areas requiring further research. Emphasize the implications of the findings for future work or practical applications. Provide a concise, insightful synthesis that ties together the central themes of the review.

Author Reply: Thank you very much for your time in studying our work. Below you will find a point-to-point response to the comments.

Comment 1: While the review presents valuable information, certain parts of the text are difficult to read. A suggestion to enhance readability and clarity is to prepare systematic tables summarizing the e.g. Raman bands discussed starting from Chapter 2. This would provide a concise reference point for the reader and facilitate a quicker understanding of key findings, trends, and assignments of the Raman spectra.

Author Reply: Thanks for such a helpful comment. We have considered that and added the table summarizing Raman band frequencies to facilitate readers in the main text of the manuscript, page (3), lines (93-99) as following:

“Raman and PL spectroscopy are effective techniques for examining many-body interactions of quasi-particles in vdW materials, including phonon anharmonicity, e-ph coupling, excitons, and phonon anisotropy, etc. Before delving through the details, we have summarized the Raman bands of the materials addressed in this review in Table 1 to provide readers with a clear overview.”

Table 1. Raman bands of materials addressed in this review.

Materials

Raman Modes

Positions (cm-1)

Origin

Figure

7-AGNRs

G

1595

in-plane C-C bond stretching vibrations

(1)

CH1

1220

the edge C-H bond bending vibrations

CH2

1255

D

1340

the edges breaking the periodicity of a perfect honeycomb lattice 

RBLM

396

the collective vibrations along the width direction of 7-AGNRs

MoO3

SVPM I

995

the stretching vibrations of Mo-O1 along [010]

(2)

SVPM II

819

the stretching vibrations of Mo-O2 along [010]

SVPM III

666

the stretching vibrations of Mo-O3 along [100]

TVPM

156

the rigid MoO4 translational vibrations

(10)

b-As

221

the coupled in-plane and out-of-plane vibrations mainly along [001] and [010]

(3, 8)

255

B2g

227

in-plane vibrations along [100]

PdTe2

Eg

74

in-plane vibrations of Te atoms in opposite directions

(4)

A1g

132

out-of-plane vibrations of Te atoms in opposite directions

Bi2Rh3Se2

233

most intense vibration but not assigned

(5, 9)

P1

192

zone-folding modes at the M2 point due to charge density wave states

P2

207

MoS2/MoO2

Ag

125

the bending vibrations of Mo-Mo bond

(11)

A1g

404

out-of-plane vibrations of S atoms of MoS2 in opposite directions

Comment 2: The included figures, while informative, are very small, making it challenging to get details. In several cases, the attempt to save space appears to compromise the figure quality and effectiveness. Larger images will improve visibility and understanding.  Ensure all axes are labeled properly with clear font, units, and descriptions for better comprehension. Use adequately sized fonts and high-resolution images to make the content legible. Review for potential typos in figure captions or labels.

Author Reply: Thank you for the suggestion. We have modified the figures quality with high-resolution images, considering clear font, units, and descriptions (Fig. 1, 2, 4, 11, ...). For example, figure 4 as following:

Figure R1, Revised Figure 4.

Comment 3: The conclusion section falls short of adequately summarizing the main findings of the review. It would be beneficial to clearly outline the key advancements in the field discussed in the review. Highlight significant gaps in knowledge or areas requiring further research. Emphasize the implications of the findings for future work or practical applications. Provide a concise, insightful synthesis that ties together the central themes of the review.

Author Reply: Thank you. The conclusion has been modified according to your helpful comment as following, please refer to the conclusion section in the manuscript;

“Many-body interaction effects ……Temperature-dependent Raman spectroscopy (TDRS) along with adopted theoretical models provide a reliable insight into the modes' thermal behaviors with temperature. For instance, the G mode of 7-AGNRs linearly redshifts with temperature, whereas the frequency shift of RBLM is nonlinearly redshifted and is mainly dominated by the 4-ph decay process. The linewidths are nonlinearly broadening, driven by the competition between 3- and 4-ph decay processes. These thermal properties of 7-AGNRs are useful for application-based AGNRs in optical and electrical devices in the future. In addition, TDRS analysis demonstrated the substrate effect on graphene (different layers) rather than the anharmonic ph-ph interaction or thermal expansion. The ph-ph scattering processes of representative SVPMs of layered α-MoO3 are primarily dominated by giant 4-ph decay, indicating its potential application in thermal devices as an emerging material with ultralow phonon thermal conductivity and high carrier mobility. TDRS of PdTe2 revealed a strong e-ph coupling based on the linewidths broadening analysis with temperature, confirming it is a phonon-mediated BCS-type superconductor, which is crucial for quantum computing applications and information processing. Furthermore, TDRS provides deep insight into the phase transitions of materials with temperature. For instance, Bi2Rh3Se2 exhibited two phases with different e-ph coupling: the 2 × 2 CCDW phase (below ~ 170 K), with zone-folding phonon modes P1, P2, and the ICCDW phase (∼170 to ∼250 K). These results indicate that Bi2Rh3Se2 may serve as a suitable material for examining the interaction between CDW and superconductivity in future work, particularly through the variations in e-ph coupling associated with the phase transitions. Different emission properties of the heterostructures (MAPbI3/MoS2 and PbI2/MoSe2) as a result of interfacial excitons are revealed by temperature-dependent PL measurements, which opens the door to the possibility of building molecularly thin heterostructures and offers a platform for studying innovative applications. The ARPR and ARPPL results showed valuable insights into the anomalous phonon polarization responses in low symmetric layered 2D materials with identify the crystallographic orientation, potentially facilitating the development of future technologies based on tunable phonon polarization, as well as engineering and controllably regulating the induced in-plane optical anisotropy in the isotropic/anisotropic heterostructures……... etc.”

Reviewer 2 Report

Comments and Suggestions for Authors The submitted manuscript is a comprehensive review of Raman and photoluminescence (PL) studies on quasiparticles in two-dimensional (2D) van der Waals (vdW) materials. ​ It focuses on the unique properties of 2D materials, such as phonon anharmonicity, electron-phonon (e-ph) coupling, exciton dynamics, and phonon anisotropy, which are crucial for their applications in electronics and optoelectronics. However, the manuscript structure and presentation can be improved for clarity and completeness.   1. Expand figure legends to explain axes, symbols, and any fitting models. 2. The discussion of temperature-dependent Raman shifts is detailed but lacks explicit connections to practical applications. 3. Some sentences are overly complex. Simplify language for better readability. 4. Clearly define specialized terms like "quasi-harmonic approximation" for a broader audience.   After the suggested changes, the manuscript could be considered for publication.

Author Response

Reviewer #2: The submitted manuscript is a comprehensive review of Raman and photoluminescence (PL) studies on quasiparticles in two-dimensional (2D) van der Waals (vdW) materials. It focuses on the unique properties of 2D materials, such as phonon anharmonicity, electron-phonon (e-ph) coupling, exciton dynamics, and phonon anisotropy, which are crucial for their applications in electronics and optoelectronics. However, the manuscript structure and presentation can be improved for clarity and completeness. 1. Expand figure legends to explain axes, symbols, and any fitting models. 2. The discussion of temperature-dependent Raman shifts is detailed but lacks explicit connections to practical applications. 3. Some sentences are overly complex. Simplify language for better readability. 4. Clearly define specialized terms like "quasi-harmonic approximation" for a broader audience. After the suggested changes, the manuscript could be considered for publication.

Author Reply: Thank you very much for your time in studying our work. Below you will find a point-to-point response to the comments.

Comment 1: Expand figure legends to explain axes, symbols, and any fitting models.

Author Reply: Thank you for your valuable feedback. We have revised the figures and their captions and resolutions to provide more detailed explanations of the axes, symbols, and any fitting models used. Please refer to the main text of the manuscript. Please refer to the reply in comment 2 to the reviewer #1.

Comment 2: The discussion of temperature-dependent Raman shifts is detailed but lacks explicit connections to practical applications

Author Reply: Thanks for the suggestion. We mentioned that in the conclusion section. Please refer to the reply in comment 3 to the reviewer #1.

Comment 3: Some sentences are overly complex. Simplify language for better readability.

Author Reply: Thank you. We have revised the manuscript to simplify the language and improve readability while maintaining the technical accuracy of the content.

Comment 4: Clearly define specialized terms like "quasi-harmonic approximation" for a broader audience.

Author Reply: Thank you for the useful suggestion; we have added the following text to the manuscript, including a clear definition and explanation of specialized terms, such as quasi-harmonic approximation: page (3-4), lines (108-115);

“The quasi-harmonic approximation is an extension of the harmonic approximation used in solid-state physics to investigate lattice vibrations (phonons). Unlike the simple harmonic approximation, in which atoms oscillate around their equilibrium positions with no interaction changes, the quasi-harmonic approximation takes into account the volume dependence of phonon frequencies, allowing for the study of thermal expansion and temperature-dependent material properties, which the harmonic approximation fails to describe accurately.”

Reviewer 3 Report

Comments and Suggestions for Authors

In this article the authors review work on many-body interactions in 2D layered materials, using their findings by Raman and Photoluminescence Spectroscopies. This work is significant regarding the .role of quasi-particles in optimizing the properties of 2D materials for electronic and optolextronic applications. 

The review is well constructed presenting the findings of the research work of the authors on this subject. 

I recommend publication as is.

Author Response

Reviewer #3: In this article the authors review work on many-body interactions in 2D layered materials, using their findings by Raman and Photoluminescence Spectroscopies. This work is significant regarding the role of quasi-particles in optimizing the properties of 2D materials for electronic and optoelectronic applications. 

The review is well constructed presenting the findings of the research work of the authors on this subject. 

I recommend publication as is.

Author Reply: Thank you very much for your positive feedback and the time you dedicated to reviewing our manuscript. We appreciate your efforts in helping us enhance the quality of our work. We are pleased to hear that the manuscript meets the criteria for publication.